# Investigation of Runoff and Sediment Yields Under Different Crop and Tillage Conditions by Field Artificial Rainfall Experiments

**Mengjing Guo [1] , Tiegang Zhang [1,2,\*], Zhanbin Li [1] and Guoce Xu [1]**

1   State Key Laboratory of Eco-hydraulics in Northwest Arid Region, Xi'an University of Technology,
    Xi'an 710048, Shaanxi, China; guomengjing263@163.com (M.G.); zhanbinli@126.com (Z.L.);
    xuguoce_x@163.com (G.X.)
2   Institute of Water Resources for Pastoral Area, Ministry of Water Resources,
    Huhhot 010020, Inner Mongolia, China
\*   Correspondence: zhang_tiegang@163.com; Tel.: +86-0471-4610326

**Abstract:** Crop types and tillage measures on slopes have significant impacts on regional water and soil conservation. In this study, we investigated the influences of multiple crop types and tillage measures on water and sediment yields based on plot-scale experiments under artificial rainfall. The objective of the study is to find the best combination of crop type and tillage measure from the perspective of reducing soil erosion. We performed artificial rainfall experiments under eight slope treatments, which are the bare-land (BL, as a reference), peanut monoculture (PL), corn monoculture (CL), bare land (upper slope) mixed with peanut monoculture (lower slope) (BP), corn and peanut intercropping (TCP), corn and soybean intercropping (TCS), downslope ridge cultivation (BS) slope, and straw-mulched (SC), respectively. Under similar rainfall intensity and initial soil moisture conditions, these treatments except for BS efficiently reduced sediment yield compared to the BL slope. In comparison, the most effective slope treatment to reduce soil erosion is TCP, followed by PL and TCS. The amount of sediment yielded from the three treatments accounts for 0.4%, 2.0%, and 3.3% of the sediment yielded from BL. We recommend the three slope treatments as the preferred choices among eight treatments. Also, the lower sediment yield in the three slope treatments benefits from their higher vegetation coverage. Vegetation coverage plays a greater role in regulating sediment yield than the surface runoff at a plot scale.

**Keywords:** field rainfall experiment; runoff generation; sediment production; cropping patterns; soil moisture

## 1. Introduction

Soil erosion is one of the main threats to the terrestrial ecosystem and is also a serious environmental issue over a large part of the world, which receives increasing attention in recent years [1–3]. Approximately 90% of the world's farmlands have suffering soil erosion [4–6]. Soil erosion brings many harms to the ecosystem and human society. First, soil erosion would decrease the fertility of the soil and lead to soil degradation, affecting soil productivity and crop yields [7–10]. Moreover, soil erosion reduces the soil water holding capacity, increasing the risk of sediment-related disasters. Also, soil erosion increases water resource pollution and sedimentation in streams and rivers, causing declines in fish and other species.

Slope farmland is an important source of soil erosion. Nearly 800 million people in the world live in slope land environments [11,12]. Slope lands are also the basic units for the occurrence and development of soil erosion in watersheds [13–15]. Therefore, reducing soil erosion on slopes is the key to soil and water conservation in the whole catchment. There are many factors affecting soil erosion process of slope lands, including rainfall intensity [10,16,17], soil types [18–21], vegetation coverage [22–26], soil moisture stages, slope, and tillage measurements. Overall, these factors can be classified into two categories: rainfall factors and underlying surface factors [27–31]. The relative importance of these factors generally varies with regions [32–35]. Therefore, it is necessary to conduct field experiments for a specific study area to analyze the dominant factor affecting the amount of soil erosion on slopes. Numerous researchers have examined the factors affecting slope soil erosion [36–40]. However, previous studies have focused on laboratory experiments [41–43], and the conclusions from laboratory experiments often do not apply to natural conditions in the field given the complexity and diversity of controlling factors on soil erosion under natural conditions.

This study area is located in the headwater of the Danjiangkou Reservoir, which is the water source area of the middle route of China's South–North Water Transfer Project and undertakes the heavy responsibility of delivering the freshwater to hundreds of millions of people in North China. However, there are a large number of slope farmlands in the reservoir area, which are the primary sources of reservoir sediment and often affect the water quality of the reservoir, particularly during the flood season (from May to October) [44]. The crop types and tillage measures of slope farmlands in the reservoir area are diverse. Different crop types and tillage measures have significant influences on soil erosion of these slope farmlands [15,24,39,45]. Therefore, scientific assessment of soil erosion under different slope cultivation practices is a better reference basis to provide scientific guildlines to the adjustment of local planting structures to reduce sediment into the reservoir. In this study, we performed artificial rainfall experiments under different crop types (corn, peanut, and soybean) and tillage measures. These crops are widely cultivated in the study area. The objective of the study is to find the best combination of crop type and tillage measure from the perspective of reducing soil erosion.

## 2. Materials and Experimental Designs

### 2.1. Study Area

The study area is located in Yingwugou watershed in Shangnan County, Shanxi Province, China and covers an area of 1.87 km$^2$. The watershed is also located in the water source area of the middle route of the South–North Water Transfer Project (see Figure 1). The water diversion project seeks to promote Northern China's economic growth by relaxing water constraints in a region now facing severe water shortage [46,47]. The watershed was selected since (1) the watershed is the region with the most serious soil erosion in the whole water source area; (2) the climatic and landscape of the watershed reflect the general characteristics of the water source area; (3) the soil and slope conditions in this watershed are suitable for growing local crops, e.g., peanut, corn, and soybean. We performed the experiments under different crop types and tillage measures. The climate of study area is the typical Monsoon-influenced humid subtropical climate. The watershed has a mean annual temperature of 14 °C and mean annual precipitation of 803.2 mm. The seasonal distribution of precipitation is very uneven throughout the year, and more than 60% of the annual precipitation occurs in wet seasons (from June to September). The primary landform type of the watershed is the rocky mountain and the elevation ranges from 464 m to 600 m a.s.l. The soil type of the watershed is dominated by yellow-brown soil and sandy loam, and the thickness of the soil layer often ranges from 20~70 cm. Slope land is an important source of food and cash crops in the study watershed due to a small number of flatlands. Common food and cash crops include corn, peanuts, soybeans, and tea trees [48].

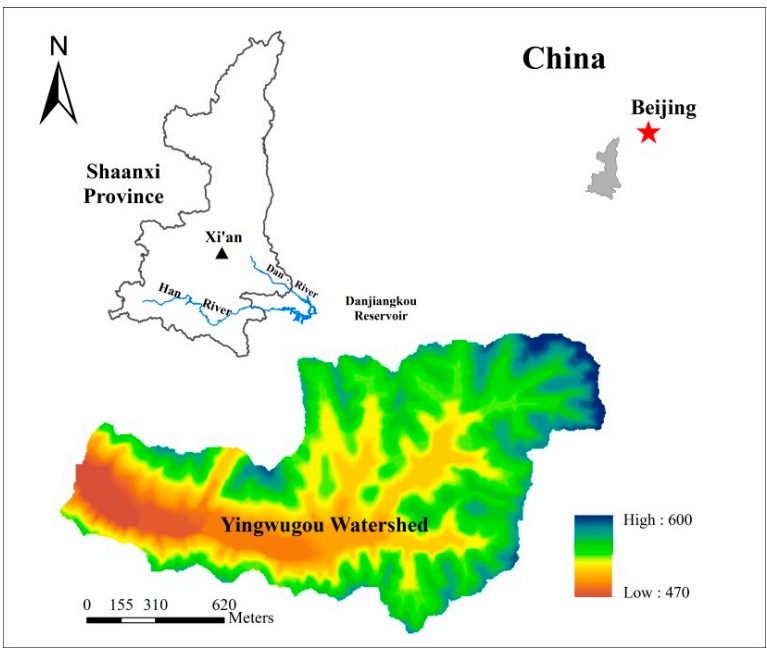

**Figure 1.** Location of the study area.

## 2.2. Experimental Design

To investigate the effects of different crop types and tillage measures on soil erosion, we designed eight slope treatments that include six land cover types and two tillage measures. Detailed procedures of artificial rainfall experiments are provided in Figure 2. The six crop cover types are the barren land (BL), peanut monoculture (PL), corn monoculture (CL), bare land (upper slope) mixed with peanut monoculture (lower slope) (BP), corn mixed with peanut intercropping (TCP), and corn mixed with soybean intercropping (TCS), respectively (see Figure 3). The two tillage measures are the downslope ridge cultivation (BS) and straw mulching (SC), respectively. The BS treatment was employed here because of the lack of irrigation facilities for the slope farmlands in the study area. Given the lower precipitation from October each year, local farmers often employ the BS slope treatment to increase survival rates of seedlings. The crop seedlings are sown between two ridges, which can gather the runoff generated from rainfall more easily and thus increase the soil moisture content at the bottom of the ridges. The BL is often used as a reference to test the soil and water conservation effects of other slope treatments. Table 1 lists the land cover types, tillage measures and crop coverage for each slope treatment. Soil bulk density was determined using the cutting ring method. The average soil bulk densities were 1.26, 1.34, and 1.42 g/cm$^3$ at depths of 10, 20, and 30 cm on the slope, respectively. The rainfall intensity for each slope treatments is 1.20 mm/min.

The experiments were conducted at the plot scale using the artificial rainfall simulator (see Figure 4a), and each runoff plot has similar soil and slope characteristics, with a length of 10 m, a width of 2 m and a slope of 10° (see Figure 4b). We employed an artificial rainfall device with downward spray nozzles to model the natural rainfall occurrence. In the early stage of equipment commissioning, we evenly arranged 100 rain buckets in the experimental plot. The rainfall uniformity degree of the device can reach more than 85%, and the rainfall intensity can be automatically controlled. The designed rainfall intensity and duration of artificial rainfall experiment are 1.2 mm/min and 60 min, respectively. During the first 10 min after the experiment started, the surface runoff and sediment yields were recorded every 2 min a plastic bucket; afterward, the runoff and sediment were collected every 5 min. The runoff the sediment yields were determined using the electric scale and pycnometer method, respectively. The moisture content of the soil on each slope was determined using the online soil moisture monitoring system (manufactured by Spectrum Technology, Aurora, IL, USA) with measurement accuracy within ±3% (25 °C). Prior to the experiment, soil moisture sensors of the online

soil moisture monitoring system were embedded in the soil on the slope. To monitor the spatial pattern of soil moisture in the runoff plot, sensor probes SW1–SW3, SW4–SW6, and SW7–SW9 recorded the volumetric moisture content (VMC) at a location 7, 5, and 3 m away from the outlet of the runoff plot at depths of 10, 20, and 30 cm, respectively (see Figure 4b).

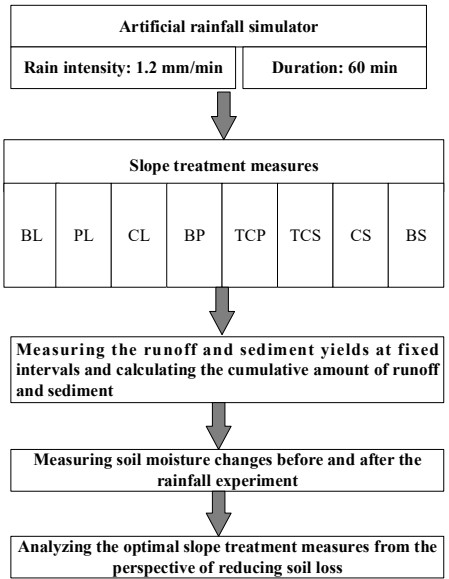

**Figure 2.** Flow chart of artificial rainfall experiments. The abbreviations in the figure denote different slope treatments, and the full name of each abbreviation is provided in Table 1.

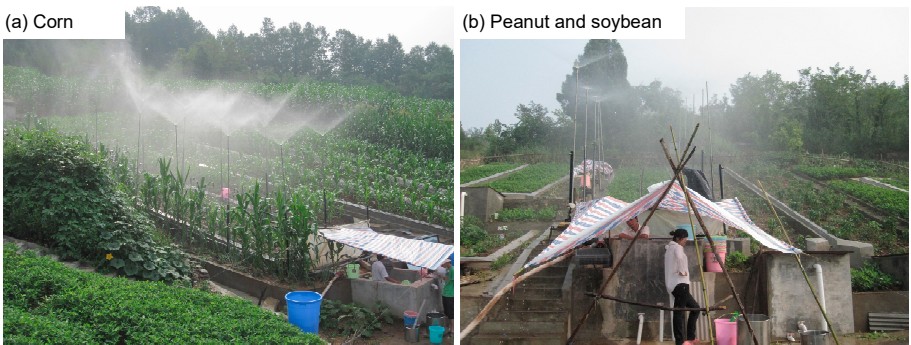

**Figure 3.** Some typical slope treatments.

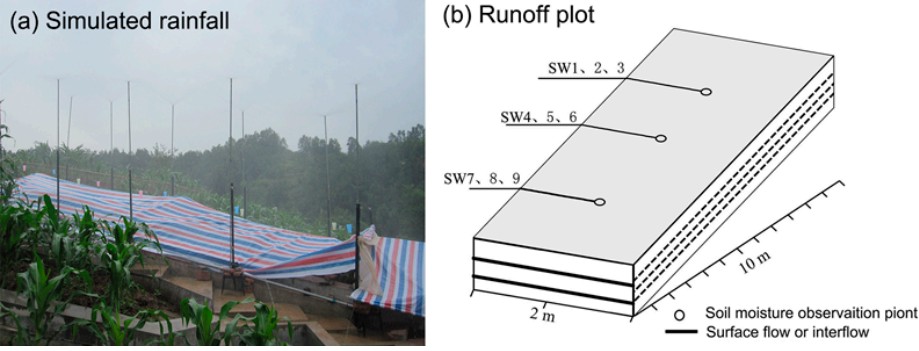

**Figure 4.** Artificial rainfall experiments on a runoff plot (**a**) and the location of soil moisture sensors in the runoff plot (**b**).

**Table 1.** Design of land cover types and tillage measures for artificial rainfall experiments

| Symbol | Name | Coverage (%) | Crop Types and Tillage Measures |
|--------|------|--------------|--------------------------------|
| BL | Bare land | 0 | Established after crop harvest |
| PL | Peanut | 92 | Crop spacing: 0.3 × 0.4 m |
| CL | Corn | 15 | Crop spacing: 1.0 × 0.5 m |
| BP | Bare land (upper slope) mixed with peanut (lower slope) | 80 | Peanuts were removed from the upper section of the slope to establish a bare land, whereas peanuts were preserved on the lower section of the slope |
| TCP | Corn mixed with peanut intercropping | 80 | Crop spacing: 0.1 × 0.5 m and (corn) 0.2 × 0.1 m (peanut) (peanut and corn were intercropped) |
| TCS | Corn mixed with soybean intercropping | 55 | Crop spacing: 0.1 × 0.5 m (corn) and 0.3 × 0.3 m (soybean) (soybean and corn were intercropped) |
| BS | Downslope ridge cultivation | 0 | After peanuts were harvested, the peanut-cropped plot was redeveloped into a bare land for downslope ridge cultivation |
| SC | Straw-mulched bare land | 75 | After peanuts were harvested, the peanut-cropped plot was mulched with straw |

## 3. Results and Discussions

### 3.1. Runoff Yield

Figure 5 shows the surface runoff production process under different slope treatments. During the initial period of the rainfall, runoff yield gradually increased with increasing rainfall duration for all treatments, but the increased rates among these treatments are different. The CL, BL, and SC slopes have a higher rate than the other slopes. During the middle period of the rainfall, the runoff yield fluctuated around a certain value. After the rainfall stopped, the runoff yield decreased rapidly.

In terms of total runoff volume, The BL slope had the highest runoff yield (195.4 L/h), followed by the CL (182.1 L/h), SC (146.7 L/h), TCS (98.7 L/h), BS (76.8 L/h), BP (54.9 L/h), PL (23.4 L/h), and TCP slopes (11.9 L/h) (see Figure 6). The runoff yields of the CL, SC, TCS, BS, BP, PL, and TCP slopes are 93, 75, 51, 39, 28, 12, and 6% of that of the BL slope, respectively. The low runoff yield for the TCP and PL slopes may benefit from that the two treatments have higher vegetation coverage (80% and 92%) than other treatments. The high vegetation coverage could retard surface runoff production, prolong the runoff residence time, and increase water infiltration into the soil, which can reduce the runoff yield significantly. Compared with the BL slope, runoff generation time on the BS slope was delayed by 33 min. This is probably because the ridge cultivation altered surface roughness and delayed the runoff production on the slope.

The crop types and tillage measures have significant influences on peak flows among these slope treatments, in which CL has the highest peak flow, followed by BL, SC, BS, TSC, BP, PL, and TCP has the smallest peak flow (see Figure 7). The time of peak flow for these slope treatments is also different from each other. The time peak flow for two slope treatments (i.e., PL and CL) occurred in the initial period of the rainfall experiment and for five treatments (e.g., BP, BL, and TCP) appeared in the late period of the experiment.

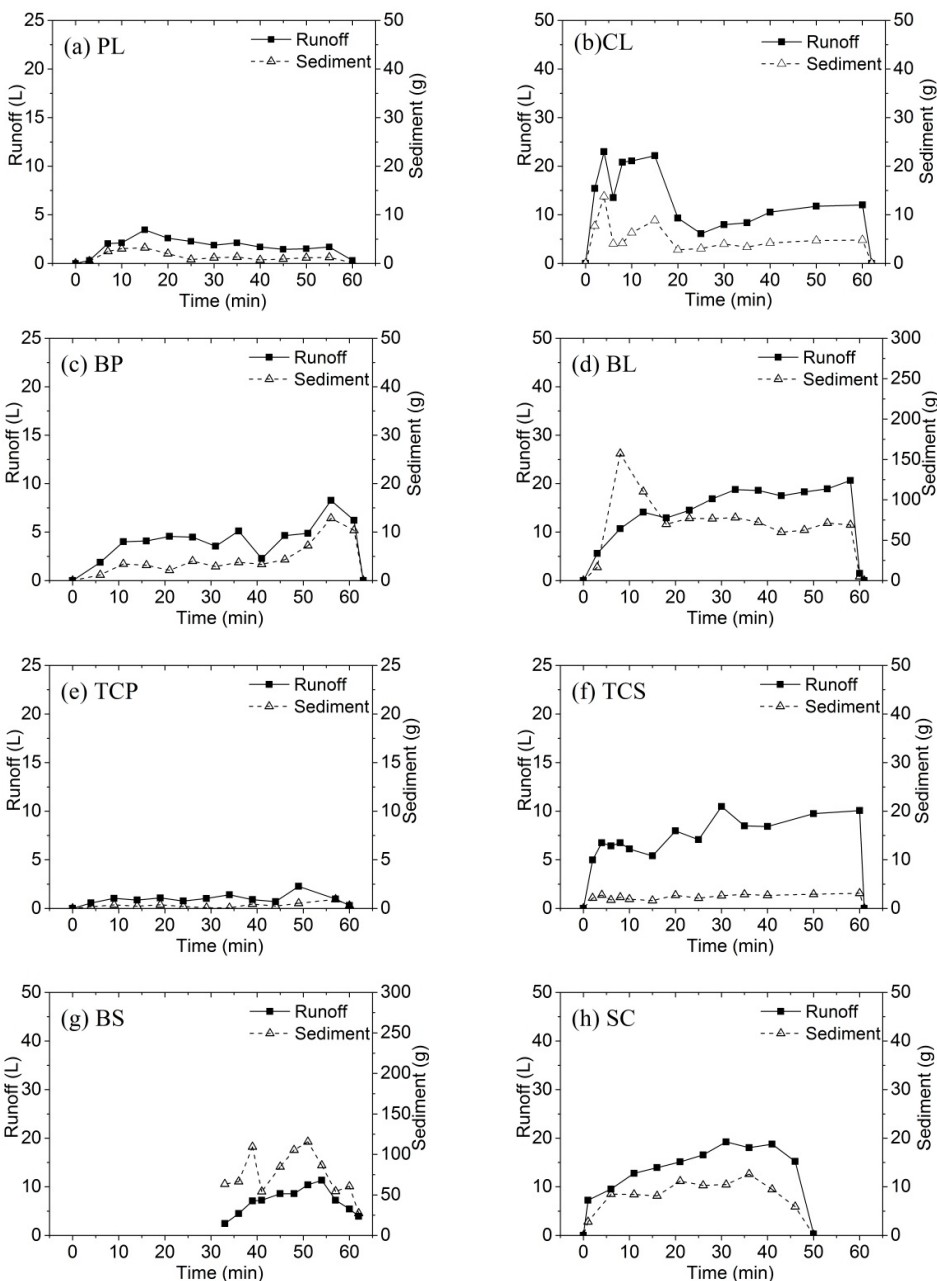

**Figure 5.** Runoff and sediment yields under different land cover and tillage conditions.

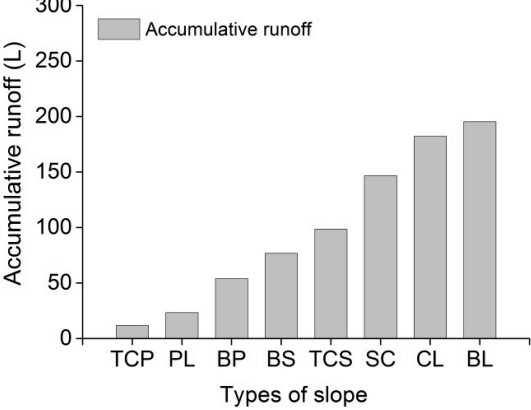

**Figure 6.** Accumulative runoff under different vegetation cover and tillage.

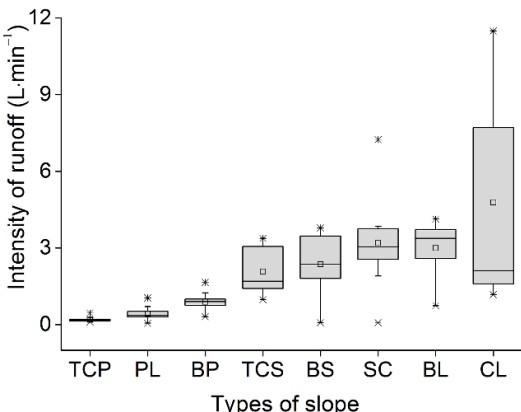

**Figure 7.** Average intensity of runoff with different slope treatments.

The reduction extents of runoffs also exhibited obvious differences between different types of crops interplanted. The runoff yields of the TCP and TCS slopes were 6% and 51% of that of the BL slope, respectively. Thus, regarding surface runoff, the intercropping of corn and peanut was more effective than the intercropping of corn and soybean. In addition to the vegetation coverage, the cultivation of a combination of tall and short crops in a crisscross pattern and suitable crop spacing can more effectively disrupt surface runoff production and reduce the runoff yield. The runoff yield of the BS slope was 39% of that of the BL slope, indicating that increasing surface roughness could retard runoff production.

Soil crusts were formed on the BL slope after the rainfall simulation experiment. When rainfall simulation was performed on the BL slope mulched by corn straw (crop coverage: 75%) (i.e., the SC slope), the combined action of soil crusts and high-coverage straw might lead to the formation of extremely favorable underlying surface conditions for runoff production, which accounted for the relatively high surface runoff yield.

Based on aforementioned analysis, although similar fluctuating patterns of runoff production were observed from slopes covered by various types of crops, the average runoff production intensity varied significantly among slopes. The slopes with higher vegetation coverage exhibit a higher capacity for runoff regulation and tend to produce a lower runoff peak flow than the slopes with low vegetation coverage (below 55%, see Table 1). The tillage measures altered surface roughness and retarded and delayed runoff production than the bare slope. Among these slope treatments, the BS and BL slopes had the highest runoff yields, and their maximum runoff yield was 16 times than that of a crop-covered slope, indicating that the capacity of the crop-covered slopes to regulate runoff was slightly higher than that of the slopes subjected to tillage only.

Here, the rainfall-runoff experiments were performed at a plot scale. To expand the experiment results to similar slope lands in other regions, we used the observed surface runoff to calibrate the unique parameter (CN, curve number) associated with runoff generation in the Soil Conservation Service Curve Number (SCS-CN) model (see Appendix A). The SCS-CN model is a simple but efficient method for determining direct runoff from storm rainfall in a particular area. It is also used as the module of runoff generation for the famous hydrological model SWAT (Soil and Water Assessment Tool). This model was developed based on empirical analysis of large amounts of rainfall-runoff data from small catchments and hill-slope plots [49,50]. Here we first converted the unit of runoff yield from L/h to mm/h. We then calibrated the parameter CN using the observed runoff yield and the least-squares method. The calibrated CN values for eight slope treatments are provided in Table 2. Caution should be exercised when applying these CN values on the catchment scale given the issue of scale mismatch. The landscape characteristics (e.g., soil, vegetation, and terrain) typically have greater spatial heterogeneity on the catchment scale than on the plot scale.

**Table 2.** Curve number in the SCS-CN model calibrated by runoff yields from artificial rainfall experiments.

| Slope Treatments | BL | CL | PL | BP | TCP | TCS | BS | SC |
|---|---|---|---|---|---|---|---|---|
| CN | 63.6 | 62.8 | 48.6 | 52.7 | 46.5 | 56.8 | 54.9 | 60.5 |

*3.2. Sediment Yield*

Figure 6 also shows the sediment yield process on each slope. The sediment production process exhibited a similar variation pattern to the runoff production process because surface runoff is the primary carrier of sediment. As for the total sediment yield, the BL slope had the highest sediment yield (946.3 g/h), followed by the BS (827.4 g/h), SC (87.9 g/h), CL (67.2 g/h), BP (58.6 g/h), TCS (31.5 g/h), PL (18.7 g/h), and TCP slopes (4.0 g/h) (see Figure 8). The sediment yields of the BS, SC, CL, BP, TCS, PL, and TCP slopes were 87.4, 9.3, 7.1, 6.2, 3.3, 2.0, and 0.4% of that of the BL slope, respectively. Regarding the intensity of sediment yield, the TCP slope had the lowest sediment production intensity, followed by the PL, BP, SC, TCS, CL, BL, and BS slopes (see Figure 9). The results demonstrate that the sediment yields of the TCS, PL, and TCP slopes were 3.3, 2.0, and 0.4% of that of the BL slope, respectively, indicating that high vegetation coverage could significantly reduce the sediment yield of the slope. In addition, compared to the BL Slope, the straw mulching was significantly efficient in trapping sediment. The sediment yield of the SC slope was only approximately 10% of that of the BL slope.

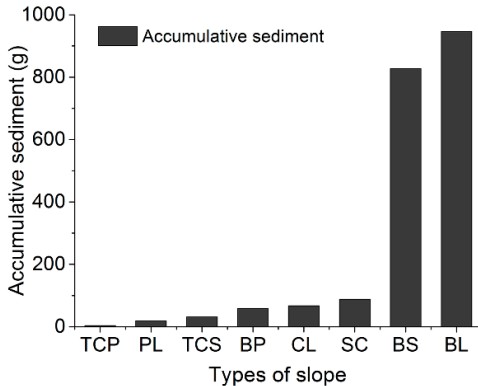

**Figure 8.** Accumulative sediment yields under different slope treatments.

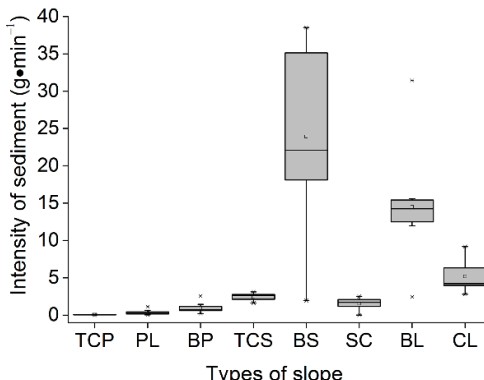

**Figure 9.** Average intensity of sediment yield with different slope treatments.

Surface runoff is the driving force of soil erosion, and the amount of surface runoff directly influences the amount of sediment production during a rainfall event [12,21]. The vegetation coverage can reduce the kinetic energy of raindrops, increase water infiltration, reduce splash erosion of the soil surface by raindrops and reduce the amount of splash-eroded surface through their leaves and stalks [23,51]. Here, we compared the relationship between sediment yield and runoff yield and vegetation coverage (see Figure 10). Results indicated that vegetation coverage plays a greater role in

regulating sediment yield than surface runoff. The coefficient of determination ($R^2$) between sediment yield and runoff yield is 0.64, while the $R^2$ between sediment yield and runoff yield is 0.17. In addition, the BS slope does not efficiently reduce soil erosion compared with the BL slope. That may because the steep slope formed from the ridges increased rill erosion. Also, the straw used to mulch the slope surface trapped rainwater, forming low-lying water accumulation areas, which caused sediment to deposit and weaken the capacity of surface runoff to transport sediment [24].

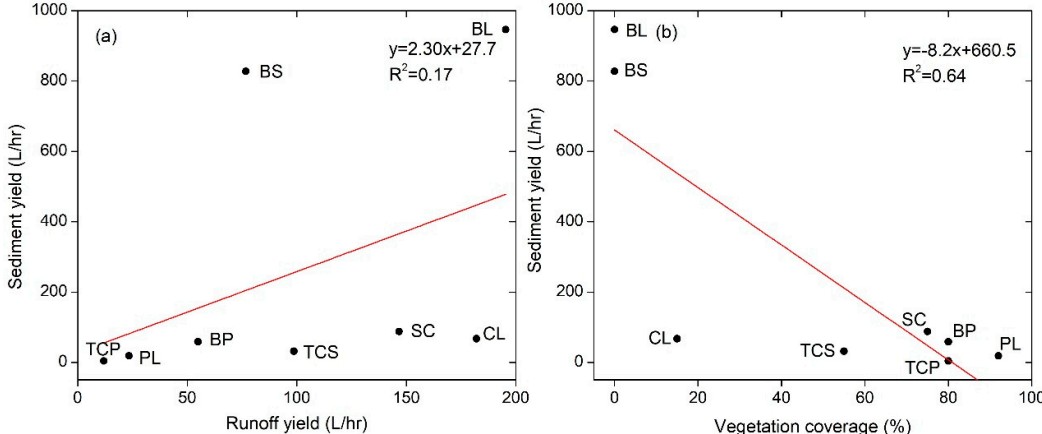

**Figure 10.** The relationship between sediment yield and runoff yield (**a**) and between sediment yield and vegetation coverage (**b**). The red lines denote the linear regression lines.

### 3.3. Changes in Volumetric Soil Moisture (VMC)

Here, we also analyzed the effects of different slope treatments on the spatial distribution of slope soil moisture during and after rainfall period. Based on the changes in the VMC of the soil recorded by the soil moisture monitoring system, five slopes (PL, BP, BL, TCP, and SC) were selected. The changes in the VMC of the soil on each slope were visually rendered using color gradient scales in ArcGIS10.1. Figure 11 shows the dynamic changes in the VMC of the soil on various slopes. After the initial rainfall, the color block representing the VMC of the topsoil turned gradually from light yellow or orange to blue or deep blue. Within the 60 min of continuous rainfall, the VMC of the soil gradually increased. At the end of rainfall (i.e., after 60 min), the color block representing the VMC of the soil on each slope turned from blue to yellow or orange, demonstrating the redistribution process of soil moisture. The VMC of the soil decreased extremely slowly and reached a relatively stable value after 8 h or an even longer time.

Table 3 shows the difference in VMC before and after a rainfall event at depths of 0–10, 10–20, and 20–30 cm. The enhancement extent of VMC of the soil after a rainfall event on the PL slope was the highest (6.3%), followed by TCP (6.1%), BP (4.3%), SC (4.0%), and BL slopes (3.1%). It means that the PL and TCP slopes had the comparable capacities in regulating the soil moisture, which was approximately twice higher than that of the BL slope. The increase in the VMC of the soil on the BP slope was 1.4 times higher than those on the BL slope, indicating that greater the vegetation coverage benefited the regulating capacity of soil moisture. There was no significant difference in the increase in the VMC of the soil between the SC and BP slopes, which indicated that straw mulching and crop cover have similar soil infiltration capacity during a rainfall event.

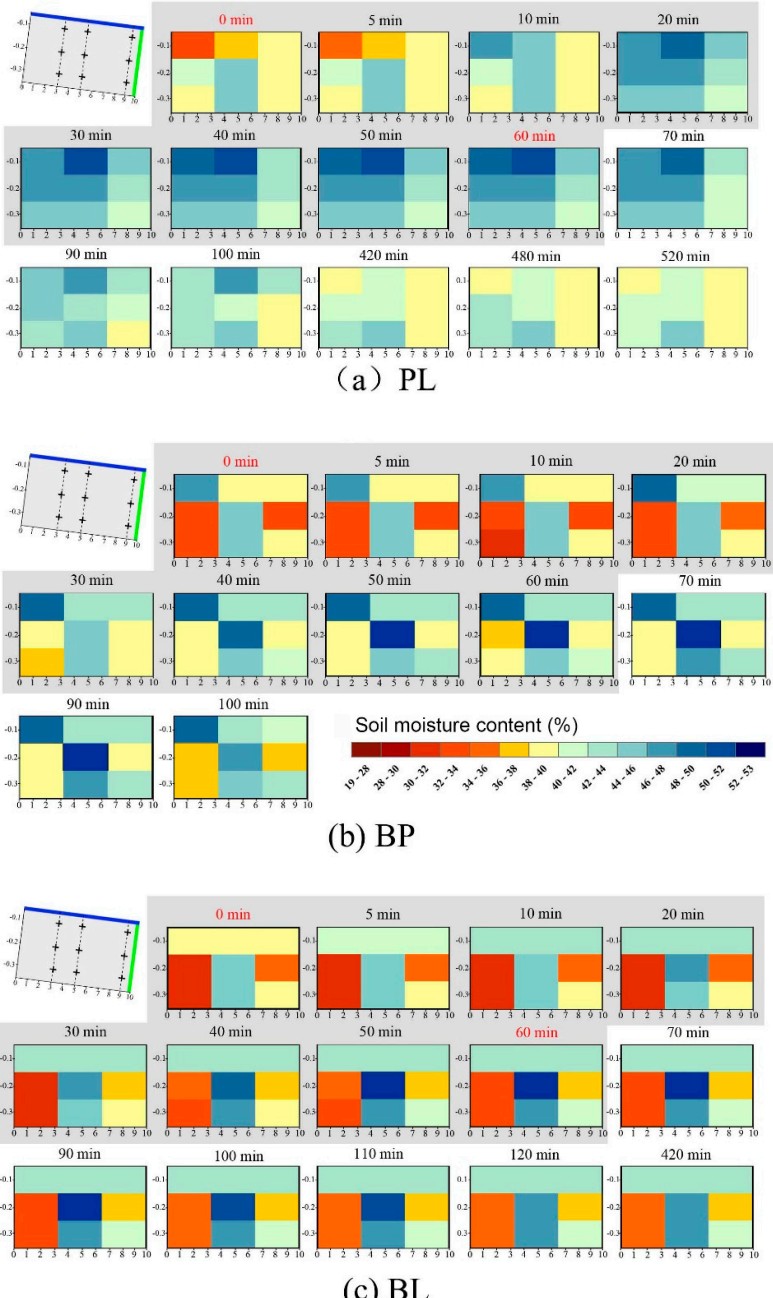

**Figure 11.** *Cont*.

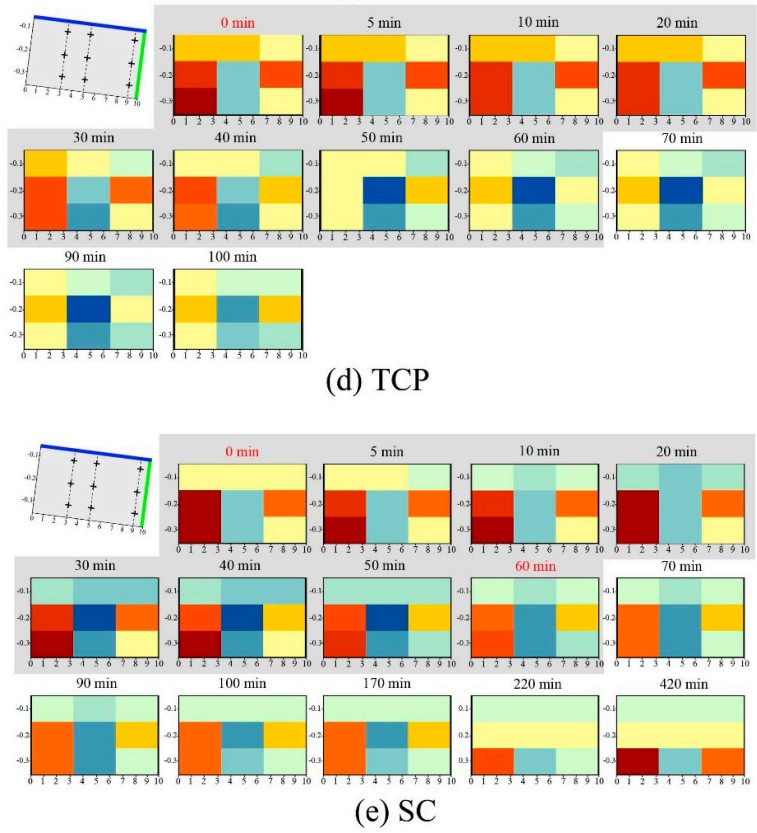

**Figure 11.** The dynamic distribution of VWC in soil profile under different vegetation cover and tillage conditions.

**Table 3.** Changes in the VWC of the soil at various depths before and after rainfall

| Number | PL | BP | BL | TCP | SC |
|---|---|---|---|---|---|
| Pre-rainfall VMC | | | | | |
| 0–10 cm | 36.4 | 41.6 | 39.5 | 33.9 | 39.5 |
| 10–20 cm | 41.4 | 36.9 | 37.6 | 28.1 | 36.9 |
| 20–30 cm | 41.2 | 38.7 | 38.4 | 36.2 | 37.7 |
| **Average VMC** | **39.7** | **39.1** | **38.5** | **32.7** | **38.0** |
| Post-rainfall VMC | | | | | |
| 0–10 cm | 48.1 | 45.2 | 43.4 | 39.5 | 44.1 |
| 10–20 cm | 45.9 | 43.0 | 41.3 | 34.9 | 41.1 |
| 20–30 cm | 43.9 | 41.9 | 40.1 | 42.1 | 40.9 |
| **Average VMC** | **46.0** | **43.4** | **41.6** | **38.8** | **42.0** |
| ΔVMC (post-rainfall VMC-pre-rainfall VMC) | | | | | |
| 0–10 cm | 11.6 | 3.6 | 3.9 | 5.6 | 4.6 |
| 10–20 cm | 4.5 | 6.1 | 3.7 | 6.8 | 4.2 |
| 20–30 cm | 2.7 | 3.2 | 1.8 | 5.9 | 3.2 |
| **Average ΔVMC** | **6.3** | **4.3** | **3.1** | **6.1** | **4.0** |

Table 4 lists the changes in the VMC of the soil at various locations before and after the rainfall. The larger difference in soil moisture (ΔVMC) before and after rainfall, the more precipitation infiltrates into the soil during the rainfall process. The PL slope has the largest ΔVMC on average, followed by TCP, BP, SC, and BL. This ranking is generally consistent with the ranking in vegetation coverage of these slope treatments (see Table 1). The results indicated that the slope treatment with larger

vegetation coverage tends to enhance soil moisture content by enhancing soil infiltration during a rainfall event.

**Table 4.** Variations of VWC in different slope positions before and after rainfall

| VMC of the Soil | Location on the Slope | PL | BP | BL | TCP | SC |
|---|---|---|---|---|---|---|
| Pre-rainfall VMC | Upper section | 38.3 | 37.7 | 34.0 | 28.7 | 32.4 |
| | Middle section | 41.9 | 42.5 | 43.2 | 27.8 | 43.4 |
| | Lower section | 38.8 | 37.0 | 38.3 | 41.8 | 38.3 |
| Post-rainfall VMC | Upper section | 46.6 | 42.0 | 36.8 | 35.4 | 36.7 |
| | Middle section | 47.8 | 47.0 | 47.4 | 33.5 | 47.5 |
| | Lower section | 43.4 | 41.1 | 40.6 | 47.5 | 41.9 |
| ΔVMC (post-rainfall VMC-pre-rainfall VMC) | Upper section | 8.3 | 4.3 | 2.8 | 6.7 | 4.3 |
| | Middle section | 5.9 | 4.5 | 4.2 | 5.7 | 4.2 |
| | Lower section | 4.6 | 4.1 | 2.3 | 5.8 | 3.6 |
| | Average | 6.3 | 4.3 | 3.1 | 6.1 | 4.0 |

## 4. Conclusions

In this study, the effects of different crop types and tillage measures on runoff and sediment production were investigated based on artificial rainfall experiments at a plot scale. Compared with bare-land (BL), corn and peanut intercropping (TCP) and peanut monoculture (PL) can reduce the runoff by 94% and 88%, respectively; the other crop types and tillage measures can reduce runoff yield by 28–75%. The measures with the largest reduction in sediment were TCP, PL, and corn and soybean intercropping (TCS), respectively. The sediment yield from the three treatments accounts for 0.4, 2.0, and 3.3% of accumulative sediment yield from BL. Thus, we recommend the three treatments as the preferred choices among these slope treatments. The specific slope treatment that farmers choose in practices needs to be based on the benefits and labor costs of these treatments. In addition, the accumulative amount of sediment mainly depends on the vegetation coverage of the slope, rather than the total surface runoff, during a rainfall event. Slopes with high vegetation coverage can effectively increase soil infiltration and moisture content, thereby reducing surface runoff and sediment yield.

**Author Contributions:** T.Z. designed and carried out the artificial rainfall experiments and analyzed the data; Z.L. and G.X. discussed the results and the first manuscript draft was written by M.G.; all authors revised the draft and approved the final manuscript.

**Funding:** This research was supported by the National Natural Science Foundations of China, grant no. 51609196, 41807156 and 41701327; the fund of the State Key Laboratory of Eco-hydraulics in Northwest Arid Region, grant no. 2018KFKT-2.

**Conflicts of Interest:** The authors declare no conflict of interest.

## Appendix A. Equation of SCS-CN Model

The general equation for the SCS-CN model is

$$\begin{cases} Q = \frac{(P-0.2S)^2}{P+0.8S} & P > 0.2S \\ Q = 0 & P \leq 0.2S \end{cases} \tag{A1}$$

$$S = \frac{25400}{CN} \tag{A2}$$

where $Q$ is the surface runoff (mm), $P$ is the precipitation (mm), $S$ is the potential maximum retention after runoff begins (mm).

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
