# Peer review of "Investigation of Runoff and Sediment Yields Under Different Crop and Tillage Conditions by Field Artificial Rainfall Experiments"

_water, doi:10.3390/w11051019_

Round 1

Reviewer 1 Report

Dear Authors,

This study aims to simulate the soil loss, runoff, soil water content and other physical quantities of eight types of soil crop and tillage types by field artificial rainfall equipment. This manuscript is an important research topic and is of great scientific value.

However, the whole manuscript is not well-structured and some sections are not clear. With regards to the methodology employed, the authors should present a series of high-resolution pictures/graphs to illustrate the observations of the experimental process, to scientifically explain the results of the observations, and to prove its research value. And, English needs to be improved for the clarity of readers.

In conclusion, this reviewer suggests that the manuscript is not acceptable in its present form, and the manuscript needs to undergo a major revision that focuses on descriptions to experimental design of artificial rainfall and discusses the abutment test results related to practical application of soil and water conservation. Finally, the detailed comments are found in the attached PDF highlighted in yellow.

Author Response

Reviewer #1:

1) This study aims to simulate the soil loss, runoff, soil water content and other physical quantities of eight types of soil crop and tillage types by field artificial rainfall equipment. This manuscript is an important research topic and is of great scientific value. However, the whole manuscript is not well-structured and some sections are not clear. With regards to the methodology employed, the authors should present a series of high-resolution pictures/graphs to illustrate the observations of the experimental process, to scientifically explain the results of the observations, and to prove its research value. And, English needs to be improved for the clarity of readers.

Answer: We have revised the corresponding section as your suggestion. We have rewritten some sections to better convey our intention to readers. We also added some pictures about the slope treatments in the revised MS, but some pictures that you required were not taken at that time. We also have polished the English writing of the MS by a professional language edit company (AJE, www.aje.com).

2) In conclusion, this reviewer suggests that the manuscript is not acceptable in its present form, and the manuscript needs to undergo a major revision that focuses on descriptions to experimental design of artificial rainfall and discusses the abutment test results related to practical application of soil and water conservation. Finally, the detailed comments are found in the attached PDF highlighted in yellow.

Answer: suggestion is taken. We have rewrote the conclusion section and deleted some repeat content in the text.

Reviewer 2 Report

The document is well written and well paced. All the elements for a good presentation are accounted for. The big concern of this reviewer si focused on the experimental methodology itself. Rainfall machines to simulate rainfall and runoff have been widely used in the las decades. My questions are the following:

-. Do the authors consider to compare the runoff results with any kind of erosion and/or rainfall runoff model? While if the experimental set up is correct the nature is simulated appropriately, the question is how reliable are those experiments. 

-. Rainfall and runoff are influenced by terrain elevation and coverage. For the study to be generalizable and useful for other areas, it would be necessary to account for those facts. Even if the crop area is small enough, it will be, as far as it seems, greater than the experimental setup area. How do the authors consider the influence of size and large scale changes in the elevation of the area, in the rainfall and run off process? While that point could be deduced up to some extent from an appropriate combination of experimental conditions, there would be still a lack of some aspects regarding the stochastic nature of the rainfall agent and its consequences on the sediment yield, that might not be apparent form the experiment. How do the authors have accounted for that?

Author Response

Reviewer 2

The document is well written and well paced. All the elements for a good presentation are accounted for. The big concern of this reviewer si focused on the experimental methodology itself. Rainfall machines to simulate rainfall and runoff have been widely used in the las decades. My questions are the following:

-. Do the authors consider to compare the runoff results with any kind of erosion and/or rainfall runoff model? While if the experimental set up is correct the nature is simulated appropriately, the question is how reliable are those experiments. 

-. Rainfall and runoff are influenced by terrain elevation and coverage. For the study to be generalizable and useful for other areas, it would be necessary to account for those facts. Even if the crop area is small enough, it will be, as far as it seems, greater than the experimental setup area. How do the authors consider the influence of size and large scale changes in the elevation of the area, in the rainfall and run off process? While that point could be deduced up to some extent from an appropriate combination of experimental conditions, there would be still a lack of some aspects regarding the stochastic nature of the rainfall agent and its consequences on the sediment yield, that might not be apparent form the experiment. How do the authors have accounted for that?

Reviewer 3 Report

This research aimed to investigate which could be the most suitable crop and tillage management to reduce soil erosion along slopes.

In the following points some suggested revisions:

- Abstract: overall, the abstract is too long. I think authors should try to reduce it consistently without losing important information. For example, I really don't understand why concluding with such a sentence "Also, straw mulching can effectively weaken the capacity of 33 surface runoff to sediment transportation"...it is an outcome or a future perspective?

- Materials and Methods: authors must better describe the kind of investigated soils. I mean that talking about "yellow-brown soils" it is not a scientific writing. You must use some international soil classification system. In particular, you can choose between Soil Taxonomy or World Reference Base.

Results and Discussion: what really lack in your paper is an analysis of variance (ANOVA). Indeed, you cannot make any comparison between different "slope treatments" without conducting a statistical test. You state that "such treatment had the highest" etc. then you talk about mean values, but in this way you make two mistakes. The first is that you cannot make any comparison without first conducting a bivariate statistical analysis. The second is that you should test whether your data is normally or not distributed. Since if this were not the case, then the use of the mean would not be advisable, and this should be replaced by the median.

Author Response

Reviewer #3:

1) Abstract: overall, the abstract is too long. I think authors should try to reduce it consistently without losing important information. For example, I really don't understand why concluding with such a sentence "Also, straw mulching can effectively weaken the capacity of 33 surface runoff to sediment transportation"...it is an outcome or a future perspective?

Answer: Suggestion is taken. We have removed these contents without losing important information.

2) Materials and Methods: authors must better describe the kind of investigated soils. I mean that talking about "yellow-brown soils" it is not a scientific writing. You must use some international soil classification system. In particular, you can choose between Soil Taxonomy or World Reference Base.

Answer: Suggestion is taken. We have revised our MS as your suggestion, please see the text that highlighted in Yellow.

3) Results and Discussion: what really lack in your paper is an analysis of variance (ANOVA). Indeed, you cannot make any comparison between different "slope treatments" without conducting a statistical test. You state that "such treatment had the highest" etc. then you talk about mean values, but in this way you make two mistakes. The first is that you cannot make any comparison without first conducting a bivariate statistical analysis. The second is that you should test whether your data is normally or not distributed. Since if this were not the case, then the use of the mean would not be advisable, and this should be replaced by the median.

Answer: Suggestion is taken. It should be noted that each slope treatment experiment were repeated three times and the results presented in the MS are the mean results of the three time experiments  

Round 2

Reviewer 1 Report

Dear Authors,

This revision has been improved significantly. However, some sections of this manuscript are still not clear due to some comments that have not been amended. Especially for the Section 2 “Materials and experimental designs”, the authors should draw a step-by-step procedure to explain the procedure more thoroughly (please see the previous reviewer’s comments in the first-round report). In conclusion, this reviewer suggests that the manuscript is not acceptable in its present form, and the manuscript needs to add the detailed procedures for the clarity of reader.

With my regards.

Author Response

Responses to reviewer’s comments: We are very grateful for your constructive comments. We have added a flowchart to illustrate the experimental instruments, procedures and purposes. Please see Figure 2 in detail. We also added the Figure 10 to analyze the dominant factor affecting sediment yield. Please see lines 230-238 for details.

Reviewer 2 Report

Two comments to the Authors' answers:

Reviewer comment: Do the authors consider to compare the runoff results with any kind of erosion and/or rainfall runoff model? While if the experimental set up is correct the nature is simulated appropriately, the question is how reliable are those experiments.

Authors' Answer:The results are based on measured data and do not require model validation.

New Reviewer's answer: I'm sorry to disagree up to some extent. I can assume the paper provides with exclusively measured data. But on the other hand, some information about how reliable the experimental results can be in order to apply them to other conditions/tillages/crop areas should be provided, or at least that point should be noticed somehow. Can the Authors assure that the results are not governed by escale effects and/or artificial rain nature? Because if they really are, then some issues might arise when others try to translate the results into their own cases. I don't mean  a (new) validation should be conducted. But further explanations about the extent of validity/applicability of the results should be provided, no matter if theyr are at plot scale.

Reviewer comment: Rainfall and runoff are influenced by terrain elevation and coverage. For the study to be generalizable and useful for other areas, it would be necessary to account for those facts. Even if the crop area is small enough, it will be, as far as it seems, greater than the experimental setup area. How do the authors consider the influence of size and large scale changes in the elevation of the area, in the rainfall and run off process? While that point could be deduced up to some extent from an appropriate combination of experimental conditions, there would be still a lack of some aspects regarding the stochastic nature of the rainfall agent and its consequences on the sediment yield, that might not be apparent form the experiment. How do the authors have accounted for that?

Answer:The runoff plots are based on the average slope characteristics of the study area. The runoff and sediment processes under different coverage and measures were compared. The results can only be used on slope scales.

New Reviewer's answer: Then that is related to the previous comment. Author's should provide with some explanations about the extent of validity of the results. In one hand they set the very problem from a general -at least regional- problem. On the other hand they limit their results to exclusively slope scales (by the way, a term that should be revised or otherwise it might result a bit confusing), hence the naturally expected generalization from any scientific research might be altered. 

Author Response

Responses to review’s comments: We are very grateful to your constructive comments. In the revised MS, we validated the validity of experiment results using a widely used runoff generation model, i.e., the Soil Conservation Service Curve Number (SCS-CN) model. This model includes a single parameter CN, which varies with crop types and tillage measures. We calibrated the parameter CN using the observed runoff yield and the least-squares method. The results indicated the SCS-CN model using the calibrated CN can perfectly reproduce the observed runoff from plot-scale artificial rainfall experiments (see author's note files). The calibrated CN values for eight slope treatments are provided in Table 2, which can help to expand the experiment results to similar slope lands in other regions. We also discussed the considerations when applying the calibrated parameter. More information can refer to the highlights in the revised MS.

Reviewer 3 Report

Can be accepted as it is.

Author Response

Thanks

Water EISSN 2073-4441 Published by MDPI AG, Basel, Switzerland RSS E-Mail Table of Contents Alert
Back to Top